# Structural basis for ligand recognition of the neuropeptide Y Y$_2$ receptor

Tingting Tang[1,2,3,8], Christin Hartig[4,8], Qiuru Chen[1,3,5], Wenli Zhao[1,2,3], Anette Kaiser[4], Xuefeng Zhang [1,2,3], Hui Zhang[1,2,3], Honge Qu[1,3,5], Cuiying Yi[1], Limin Ma[2], Shuo Han [2], Qiang Zhao[2,3,6,7 ✉], Annette G. Beck-Sickinger [4 ✉] & Beili Wu [1,3,5,6,7 ✉]

The human neuropeptide Y (NPY) Y$_2$ receptor (Y$_2$R) plays essential roles in food intake, bone formation and mood regulation, and has been considered an important drug target for obesity and anxiety. However, development of drugs targeting Y$_2$R remains challenging with no success in clinical application yet. Here, we report the crystal structure of Y$_2$R bound to a selective antagonist JNJ-31020028 at 2.8 Å resolution. The structure reveals molecular details of the ligand-binding mode of Y$_2$R. Combined with mutagenesis studies, the Y$_2$R structure provides insights into key factors that define antagonistic activity of diverse antagonists. Comparison with the previously determined antagonist-bound Y$_1$R structures identified receptor-ligand interactions that play different roles in modulating receptor activation and mediating ligand selectivity. These findings deepen our understanding about molecular mechanisms of ligand recognition and subtype specificity of NPY receptors, and would enable structure-based drug design.

[1] CAS Key Laboratory of Receptor Research, Shanghai Institute of Materia Medica, Chinese Academy of Sciences, 555 Zuchongzhi Road, Pudong, Shanghai 201203, China. [2] State Key Laboratory of Drug Research, Shanghai Institute of Materia Medica, Chinese Academy of Sciences, 555 Zuchongzhi Road, Pudong, Shanghai 201203, China. [3] University of Chinese Academy of Sciences, No. 19A Yuquan Road, Beijing 100049, China. [4] Institute of Biochemistry, Faculty of Life Sciences, Leipzig University, Brüderstr. 34, D 04103 Leipzig, Germany. [5] School of Life Science and Technology, ShanghaiTech University, 393 Hua Xia Zhong Road, Shanghai 201210, China. [6] CAS Center for Excellence in Biomacromolecules, Chinese Academy of Sciences, Beijing 100101, China. [7] School of Pharmaceutical Science and Technology, Hangzhou Institute for Advanced Study, UCAS, Hangzhou 310024, China. [8] These authors contributed equally: Tingting Tang, Christin Hartig. ✉email: zhaoq@simm.ac.cn; abeck-sickinger@uni-leipzig.de; beiliwu@simm.ac.cn

The human NPY receptors including four subtypes, namely $Y_1$, $Y_2$, $Y_4$, and $Y_5$ receptors, are widely distributed in central and peripheral nervous systems, as well as a variety of tissues and cell types[1,2]. In response to three endogenous peptide ligands, NPY, peptide YY, and pancreatic polypeptide[3–5], NPY receptors play important roles in a variety of physiological processes, including food intake, angiogenesis, bone formation, and regulation of circadian rhythm and mood disorder[6–10]. Therefore, NPY receptors have been proposed as important drug targets for the treatment of obesity, anxiety, cancer, and cardiovascular diseases[11,12]. However, drugs that target NPY receptors are not currently available, partly due to the poor understanding of receptor–ligand interactions. Previous studies using mutagenesis, computational modeling, nuclear magnetic resonance, and various functional assays offered insights into ligand-binding modes of NPY receptors[13–17]. In addition, crystal structures of $Y_1$ receptor ($Y_1$R) bound to two structurally diverse antagonists were recently determined, providing molecular details of ligand recognition and selectivity of a NPY receptor[13]. However, more structural information is essential to fully understand the molecular basis of ligand recognition and subtype specificity for the complex multiligand/multireceptor system of the NPY-Y receptor family. $Y_2$R has attracted considerable interest as a drug target for its role in food intake and bone formation[18–20]. A number of $Y_2$R agonists and antagonists have shown therapeutic potential in the treatment of obesity and anxiety[21–23], but their clinical application has been limited by low potency and selectivity and poor blood–brain-barrier permeability[12,18,23,24]. JNJ-31020028 is a potent, selective, brain penetrant small-molecule antagonist of $Y_2$R, and has been suggested as a potential treatment for the negative affective states following alcohol withdrawal[22,25].

In this work, we report the crystal structure of $Y_2$R in complex with JNJ-31020028 at 2.8 Å resolution. Together with extensive functional studies, our results provide key insights into the structural basis of $Y_2$R ligand-binding mode and NPY receptor subtype specificity.

## Results

**Structure determination of $Y_2$R–JNJ-31020028 complex.** To obtain diffraction-quality crystals of $Y_2$R–JNJ-31020028, an engineered $Y_2$R construct was designed by truncating 28 amino acids (S354-V381) at C terminus and introducing two mutations $H149^{3.51}Y$ and $S280^{6.47}C$ (superscripts indicate nomenclature according to Ballesteros–Weinstein numbering system[26]) to improve protein yield, homogeneity, and stability. Crystallization was further facilitated by fusing a modified T4 lysozyme (T4L)[27] at the N terminus of the receptor and replacing six residues (S251-N256) in the third intracellular loop (ICL3) with a modified flavodoxin[27]. Functional assays indicate that these modifications have little effect on binding and antagonistic activity of JNJ-31020028 and receptor signaling (Supplementary Fig. 1a–c). The $Y_2$R–JNJ-31020028 complex was obtained by copurifying the modified $Y_2$R with JNJ-31020028. The complex structure was determined at 2.8 Å resolution (Supplementary Table 1). The N-terminal T4L fusion protein was not traced due to poor electron densities. The ligand JNJ-31020028 used in protein purification is a racemic mixture of R-isomer and S-isomer (molar ratio = 1:1), which have similar $Y_2$R affinity[28]. Strong and unambiguous electron densities are present for JNJ-31020028 in the $Y_2$R structure with the S-isomer fitting better compared to the R-isomer (Supplementary Fig. 1d–g). The following structural analysis is focused on the binding mode of the S-isomer.

**Overall architecture of $Y_2$R.** The $Y_2$R–JNJ-31020028 structure exhibits a canonical seven-transmembrane helical bundle (helices

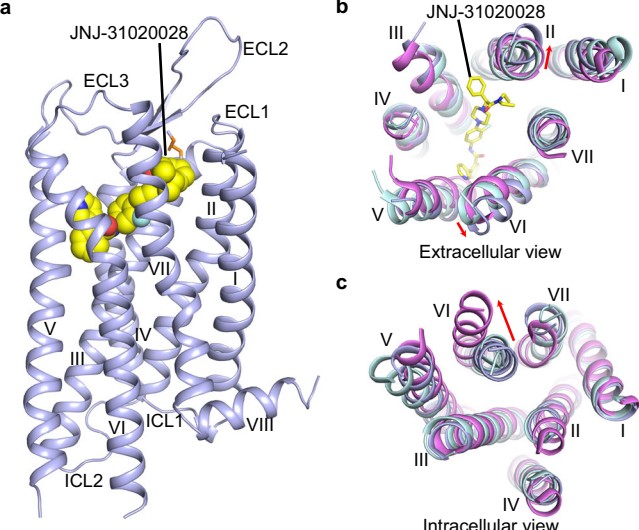

**Fig. 1 Overall structure of $Y_2$R–JNJ-31020028 complex. a** Side view of the $Y_2$R–JNJ-31020028 structure. $Y_2$R is shown in light blue cartoon representation. JNJ-31020028 (carbon in yellow, nitrogen in blue, oxygen in red, fluorine in cyan) is shown in sphere representation. The disulfide bond is shown as orange sticks. **b**, **c** Structural comparison of $Y_2$R with inactive $Y_1$R (PDB code: 5ZBQ) and active NTSR1 (PDB code: 6OS9). The helical bundles of $Y_2$R, $Y_1$R, and NTSR1 are colored light blue, light cyan, and pink, respectively. JNJ-31020028 is shown as sticks. **b** Extracellular view. Red arrows indicate the movements of helices II and VI in the $Y_2$R structure compared to the structures of $Y_1$R and NTSR1. **c** Intracellular view. Red arrow indicates the conformational change of helix VI in the active NTSR1 structure relative to the structures of $Y_1$R and $Y_2$R.

I–VII) architecture of G protein-coupled receptors (GPCRs) (Fig. 1). The second extracellular loop (ECL2) of the receptor adopts a β-hairpin conformation, which is a common structural feature shared by class A peptide GPCRs. This β-hairpin structure, together with the conserved disulfide bond connecting helix III and ECL2[29], stabilizes the conformation of the extracellular part of $Y_2$R (Fig. 1a). $Y_2$R is structurally similar to $Y_1$R (PDB code: 5ZBQ)[13], with a $C_\alpha$ root-mean-square deviation of 0.8 Å within the helical bundle. Compared to the structures of inactive $Y_1$R[13] and active neurotensin receptor 1 (NTSR1) (PDB code: 6OS9)[30], the extracellular tips of helices II and VI in the JNJ-31020028-bound $Y_2$R structure move outward by 3.6 and 2.0 Å, respectively (Fig. 1b). This movement may be partially due to the ligand binding as JNJ-31020028 would form spatial clashes with these two helices if they were in similar conformations to those in $Y_1$R and NTSR1. On the intracellular side, helix VI of $Y_2$R adopts an inward conformation similar to that observed in the inactive $Y_1$R structure but not in the active NTSR1 structure, suggesting an inactive conformational state of the $Y_2$R–JNJ-31020028 structure (Fig. 1c).

**Binding mode of JNJ-31020028 in $Y_2$R.** The $Y_2$R–JNJ-31020028 structure reveals a ligand-binding pocket formed by residues from the first extracellular loop (ECL1) and helices II–VII of $Y_2$R (Fig. 2). The binding cavity is in a depth similar to that in some known class A peptide GPCR structures, such as angiotensin receptor $AT_1R$[31], orexin receptor $OX_1R$[32], and κ-opioid receptor[33] (Supplementary Fig. 2).

The antagonist JNJ-31020028 is comprised of six functional groups, including phenylethyl, diethyl amide, benzamide, and pyridine moieties at both ends of the compound, and

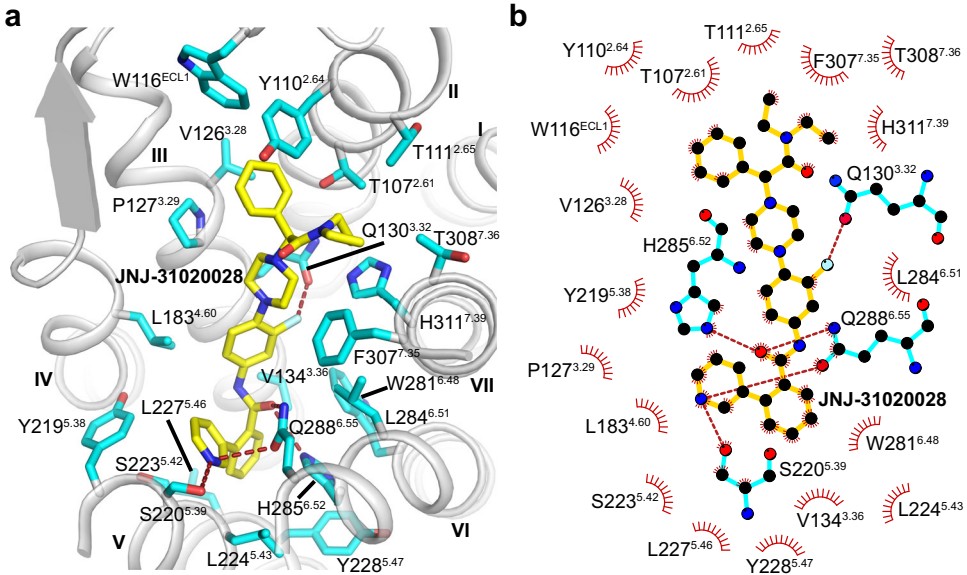

**Fig. 2 Binding mode of JNJ-31020028 in Y$_2$R. a** Ligand-binding pocket for JNJ-31020028. Y$_2$R is shown in gray cartoon representation. The Y$_2$R residues (carbon in cyan) that form interactions with JNJ-31020028 are shown as sticks. JNJ-31020028 (carbon in yellow) is shown as sticks and hydrogen bonds are shown as red dashed lines. **b** Schematic representation of interactions between Y$_2$R and JNJ-31020028 analyzed using LigPlot$^+$ program[47]. The Y$_2$R residues engaged in hydrogen bonds are shown as cyan sticks. Hydrogen bonds are shown as red dashed lines.

fluorophenyl and piperazine moieties in the middle[28]. The phenylethyl and diethyl amide groups of JNJ-31020028 bind to a cavity shaped by ECL1 and helices II, III, and VII, making hydrophobic contacts with residues T107$^{2.61}$, Y110$^{2.64}$, T111$^{2.65}$, W116$^{ECL1}$, V126$^{3.28}$, F307$^{7.35}$, T308$^{7.36}$, and H311$^{7.39}$ (Fig. 2). This aligns well with previous structure–activity relationship (SAR) studies showing that substitution of the diethyl amide group in JNJ-31020028 with ethyl amide or methyl ester decreases its binding affinity to Y$_2$R by sevenfold[28], suggesting that hydrophobicity is beneficial in this region. The importance of these hydrophobic interactions in ligand binding was also reflected by a 3–11-fold reduction in the antagonistic effect of JNJ-31020028 on inhibiting NPY-induced inositol phosphate (IP) accumulation for the Y$_2$R mutants Y110$^{2.64}$A, W116$^{ECL1}$A, V126$^{3.28}$A, F307$^{7.35}$A, and F307$^{7.35}$E (Fig. 3a, Supplementary Figs. 3 and 4a–f, and Supplementary Table 2). The involvement of the residues Y110$^{2.64}$ and F307$^{7.35}$ in ligand binding has also been reported for several other Y$_2$R antagonists such as BIIE0246[15,16], suggesting that different antagonists may share a similar binding site in Y$_2$R.

The benzamide and pyridine groups of JNJ-31020028 sit in a subpocket bordered by helices III, V, and VI, forming hydrophobic interactions with residues V134$^{3.36}$, Y219$^{5.38}$, S220$^{5.39}$, S223$^{5.42}$, L224$^{5.43}$, L227$^{5.46}$, Y228$^{5.47}$, and H285$^{6.52}$ (Fig. 2). The importance of these hydrophobic contacts for ligand binding is supported by previous SAR studies, showing that the JNJ-31020028 analogs containing the less bulky 2-ethylbutane substituent were less active than the derivatives with biphenyl or 3-phenylpyridine substituent[28]. In addition to the hydrophobic contacts, the carbonyl of the benzamide group and the nitrogen within the pyridine group form a hydrogen bond network with the residues S220$^{5.39}$, H285$^{6.52}$, and Q288$^{6.55}$ in Y$_2$R (Fig. 2). Consistent with the above interactions, the mutations V134$^{3.36}$A, Y219$^{5.38}$A, S220$^{5.39}$A, S223$^{5.42}$A, L224$^{5.43}$A, L227$^{5.46}$A, Y228$^{5.47}$A, H285$^{6.52}$T, and Q288$^{6.55}$A resulted in reduced antagonistic activity of JNJ-31020028 in the IP accumulation assay (Fig. 3a, Supplementary Figs. 3 and 4g–o, and Supplementary Table 2). Among these mutations, H285$^{6.52}$T and Q288$^{6.55}$A

displayed the largest effect, reducing the inhibitory activity of the antagonist by 20-fold and 65-fold, respectively. These data indicate that the polar network between JNJ-31020028 and the receptor helix VI plays a crucial role in mediating receptor–ligand recognition.

In the middle region of the ligand, the fluorophenyl and piperazine groups insert into a binding crevice with helices III and IV on one side and helices VI and VII on the other side. These two aromatic rings form multiple hydrophobic interactions with residues P127$^{3.29}$, Q130$^{3.32}$, L183$^{4.60}$, W281$^{6.48}$, L284$^{6.51}$, F307$^{7.35}$, and H311$^{7.39}$ in Y$_2$R (Fig. 2). The roles of these residues in ligand recognition have been studied using mutagenesis and IP accumulation assays. It was observed that mutations L183$^{4.60}$A, W281$^{6.48}$T, L284$^{6.51}$A, F307$^{7.35}$A, and F307$^{7.35}$E were accompanied with decreased antagonistic activity of JNJ-31020028, with W281$^{6.48}$T showing the most dramatic drop (186-fold) (Fig. 3a, Supplementary Fig. 4e, f, p–r, and Supplementary Table 2). The residue W$^{6.48}$ has been found to be highly important for the NPY-induced G protein activation[34]. Therefore, it is not surprising that the interaction of this residue with the antagonist, which may stabilize the receptor in an inactive conformation, is critical for suppression of the receptor activity. In addition to the hydrophobic interactions, the fluorine atom in the central phenyl core forms a hydrogen bond with the side chain of Q130$^{3.32}$. The importance of this atom in retaining high binding affinity to Y$_2$R has been shown in previous SAR studies, in which the fluoro-substituted analogs provide a significant improvement in Y$_2$R binding affinity over their unsubstituted counterparts[28]. The residue Q$^{3.32}$ is well known to be important for the interaction with the amidated C terminus of NPY[14]. This is also supported by the present work showing greatly impaired NPY-induced receptor signaling for the Y$_2$R mutant Q130$^{3.32}$H (132-fold reduction of EC$_{50}$) (Supplementary Table 2). Thus, the antagonist most likely interacts with this residue to suppress receptor functionality. Similar as we previously observed for the interaction between Y$_1$R and its antagonist UR-MK299[13], mutagenesis studies revealed an increased antagonistic activity of JNJ-31020028 when introduced the Q130$^{3.32}$H mutation in Y$_2$R

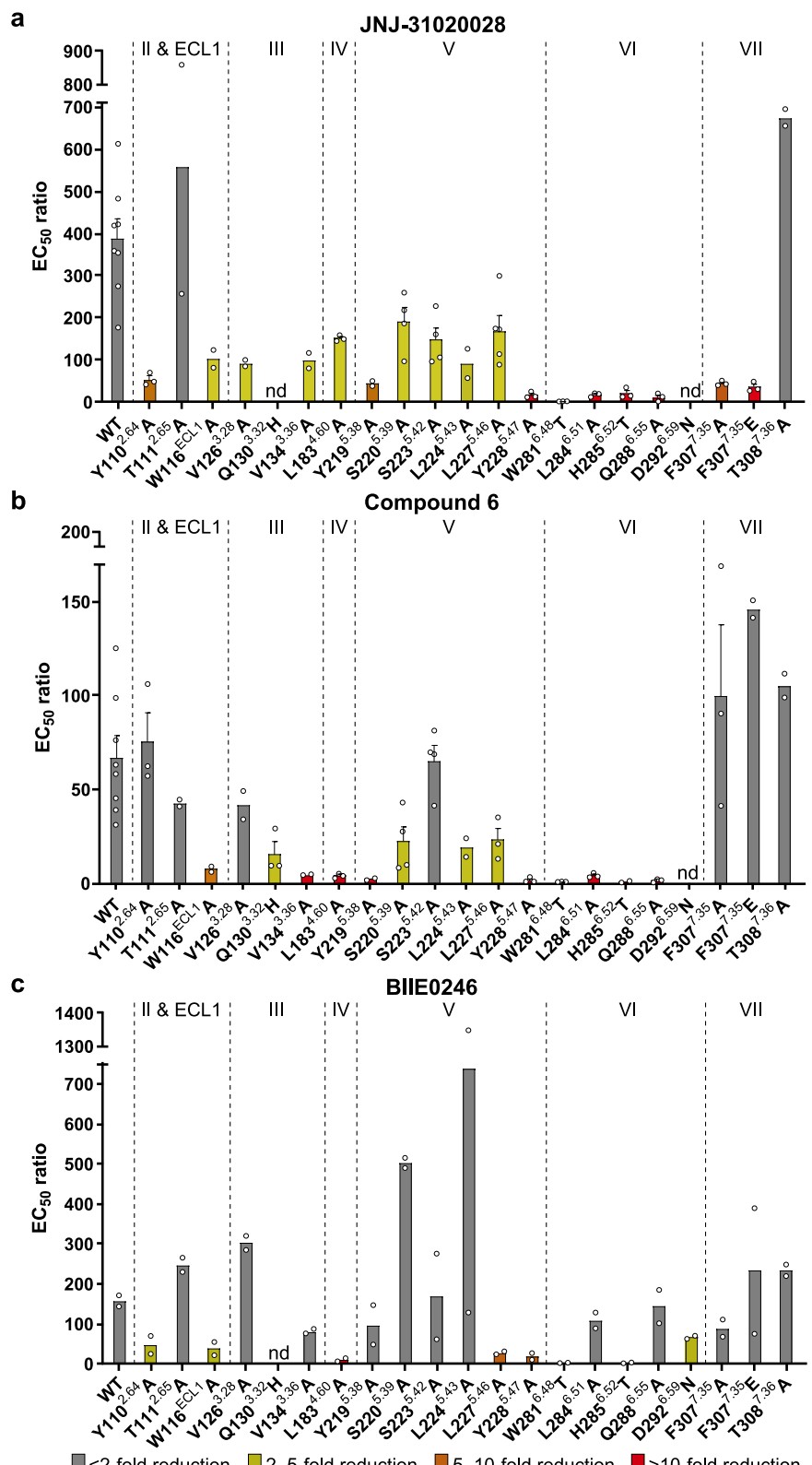

**Recognition between Y₂R and other antagonists**. To obtain more molecular details that govern ligand recognition and

(Supplementary Fig. 4s), which is likely to retain the hydrogen bond with the fluorophenyl group of the antagonist and may also introduce an additional π-stacking interaction with the ligand.

antagonistic activity, which would facilitate future drug discovery, we performed additional mutagenesis studies on two other representative antagonists of Y₂R that differ in size, structure, antagonistic activity, and blood–brain-barrier permeability[24], BIIE0246 and compound 6 (Supplementary Fig. 1h, i). By comparing the EC₅₀ ratios (EC₅₀(NPY + antagonist)/EC₅₀(NPY)) of the three different antagonists at the same concentration (1 μM) at

**Fig. 3 NPY-induced IP accumulation inhibited by antagonists. a** JNJ-31020028. **b** Compound 6. **c** BIIE0246. Bars represent $EC_{50}$ ratios of the mutated receptors compared to the $EC_{50}$ ratio of the wild-type $Y_2R$ using 1 μM concentration of the respective antagonist. The $EC_{50}$ ratio refers to the shift between the NPY and NPY + 1 μM antagonist curve ($EC_{50(NPY + antagonist)}/EC_{50(NPY)}$) and characterizes the antagonistic effect on the wild-type receptor or receptor mutants. By comparison of $EC_{50}$ ratios between wild-type and mutant receptors, influences of all tested residues on antagonistic activity were determined. A higher ratio indicates higher antagonistic activity. A reduced $EC_{50}$ ratio of mutant compared to the wild-type receptor was interpreted as important for the respective antagonist. At least two independent experiments were performed in triplicate. Where more than two experiments were performed, data are displayed as mean ± SEM (bars) with individual data points shown (dots). Where two experiments were performed, data are displayed as mean (bars) with individual data points shown (dots). Bars are colored based on the extent of effect (gray, <2-fold reduction of $EC_{50}$ ratio; yellow, 2–5-fold reduction of $EC_{50}$ ratio; orange, 5–10-fold reduction of $EC_{50}$ ratio; red, >10-fold reduction of $EC_{50}$ ratio). nd, not determined. See Supplementary Table 2 for detailed statistical evaluation. Source data are provided as a Source Data file.

the wild-type (WT) receptor, it is noticeable that JNJ-31020028 had the highest antagonistic activity as displayed by the highest $EC_{50}$ ratio ($EC_{50}$ ratio = 371). BIIE0246 showed a comparably weaker activity ($EC_{50}$ ratio = 159) and compound 6 exhibited the lowest activity as deduced from the lowest $EC_{50}$ ratio ($EC_{50}$ ratio = 61). The mutagenesis data revealed several highly relevant positions for binding of all three antagonists, including $W116^{ECL1}$, $V134^{3.36}$, $L183^{4.60}$, $L227^{5.46}$, $Y228^{5.47}$, $W281^{6.48}$, and $H285^{6.52}$ (Fig. 3a–c, Supplementary Fig. 4c, g, l–n, p, q, and Supplementary Table 2). Among these residues, $W281^{6.48}$ and $H285^{6.52}$ displayed the largest effect on the inhibitory effect of the antagonists, with their mutants $W281^{6.48}T$ and $H285^{6.52}T$ showing 48–186-fold and 20–153-fold drop of antagonistic activity, respectively. These results are consistent with the fact that helix VI exhibits the most profound conformational change during receptor activation. The interactions between the antagonists and these two aromatic residues may play a role in constraining the conformational rearrangement of receptor helix VI and thus stabilizing the receptor in an inactive state. Furthermore, most of the above residues (five out of seven) locate at the bottom of the ligand-binding pocket shaped by helices III, V, and VI, suggesting that the receptor–ligand interactions in this region are key for different antagonists to modulate receptor activation.

In the $Y_2R$–JNJ-31020028 structure, the benzamide and pyridine groups of the ligand form extensive contacts with helices V and VI of the receptor. This was supported by the impaired antagonistic activity of JNJ-31020028 when mutations were introduced to break any of the interactions (Fig. 3a and Supplementary Table 2). Similar results were also observed for compound 6 (Fig. 3b and Supplementary Table 2). However, except for the residues at the bottom of the ligand-binding cavity, the mutants of all other residues in these two helices within the binding pocket, $Y219^{5.38}$, $S220^{5.39}$, $S223^{5.42}$, $L224^{5.43}$, $L284^{6.51}$, and $Q288^{6.55}$, showed little effect on the activity of BIIE0246 (Fig. 3c, Supplementary Fig. 4h–k, o, r, and Supplementary Table 2), suggesting that this antagonist makes much more limited contacts with helices V and VI than JNJ-31020028 and compound 6. In contrast to this observation, mutation $D292^{6.59}N$ resulted in a twofold reduction of antagonistic activity of BIIE0246 (Fig. 3c, Supplementary Fig. 4t, and Supplementary Table 2). Although $EC_{50}$ ratios for $D292^{6.59}N$ could not be determined as the NPY curves in the presence of 1 μM antagonist did not reach saturation for JNJ-31020028 and compound 6 (Supplementary Fig. 4t), this residue is unlikely relevant for JNJ-31020028 due to its remote location from the antagonist in the crystal structure. Similar results were also obtained in previous investigation of the $D292^{6.59}N$ mutant of $Y_2R$, showing a loss of antagonistic activity only for BIIE0246 that contains a positively charged group, but not for the uncharged compounds 40 and 46[16]. The residue $D^{6.59}$ has been suggested to be highly important for all NPY receptors binding to NPY through an ionic interaction with one of the two arginine residues at the peptide

C terminus[35]. The importance of this acidic residue is reflected by a drastically impaired NPY potency in triggering $Y_2R$ signaling in the present study (233-fold reduction of $EC_{50}$) (Supplementary Table 2). Together, these data suggest that the ionic interaction contributed by the residue $D^{6.59}$ is key for recognition of the ligands with a positive charge.

The binding of the antagonists differs greatly in contributions of residues within helices II, III, and VII. Residue $Y110^{2.64}$ is known to be crucial for binding of the peptide agonist NPY[36]. This was also confirmed by the detected $EC_{50}$ value of NPY for the mutant $Y110^{2.64}A$ (1.8 nM), which is 30-fold increased compared to the $EC_{50}$ value at the WT receptor (0.06 nM) (Supplementary Fig. 4b and Supplementary Table 2). Inhibited by the antagonists, the $Y110^{2.64}A$ mutant displayed a reduced $EC_{50}$ ratio for JNJ-31020028 ($EC_{50}$ ratio = 52) and BIIE0246 ($EC_{50}$ ratio = 43), indicating that the antagonistic activity of these two antagonists was dropped by sevenfold and fourfold, respectively (Fig. 3a, c, Supplementary Fig. 4b, and Supplementary Table 2). By contrast, this variant showed an $EC_{50}$ ratio comparable to the WT receptor for compound 6 (Fig. 3b, Supplementary Fig. 4b, and Supplementary Table 2). In helix III, the alanine replacement of $V126^{3.28}$ that decreased the JNJ-31020028 activity by fourfold had little effect on the other two antagonists (Fig. 3a–c, Supplementary Fig. 4d, and Supplementary Table 2). In addition, the $Q130^{3.32}H$ mutant, which improved the inhibitory activity of JNJ-31020028 by potentially providing an additional π-stacking interaction with the ligand, exhibited a similar effect on BIIE0246, but led to a fourfold reduction of the activity of compound 6 (Fig. 3a–c, Supplementary Fig. 4s, and Supplementary Table 2). Diverse results for the three antagonists were also obtained for the residue $F307^{7.35}$ in helix VII. Its variants $F307^{7.35}A$ and $F307^{7.35}E$ impaired the antagonistic activity of JNJ-31020028 by 8-fold and 11-fold, respectively, while in contrast, had $EC_{50}$ ratios comparable or increased compared to the WT $EC_{50}$ ratio for BIIE0246 and compound 6 (Fig. 3a–c, Supplementary Fig. 4e, f, and Supplementary Table 2). These data suggest that these structurally diverse antagonists may share a similar binding site in $Y_2R$ but adopt different molecular patterns in interaction with the receptor.

## Discussion

JNJ-31020028 is a selective $Y_2R$ antagonist with an over 100-fold higher binding affinity to $Y_2R$ over the other NPY receptors[25]. Comparison of the $Y_2R$–JNJ-31020028 structure and the previously determined $Y_1R$ structures reveals ligand-binding pockets differing in shape (Fig. 4a–c). JNJ-31020028 and the $Y_1R$ antagonists UR-MK299 and BMS-193885 share a similar binding site at the bottom of the ligand-binding cavity bordered by helices III–VI (Fig. 4d, e). In contrast, the receptor–ligand interaction modes in $Y_1R$ and $Y_2R$ are substantially different on the extracellular side of the binding pocket. In $Y_1R$, UR-MK299 and BMS-193885 extend upward along helices VI and VII to engage interactions with the key residue $D^{6.59}$ and the residues around[13]

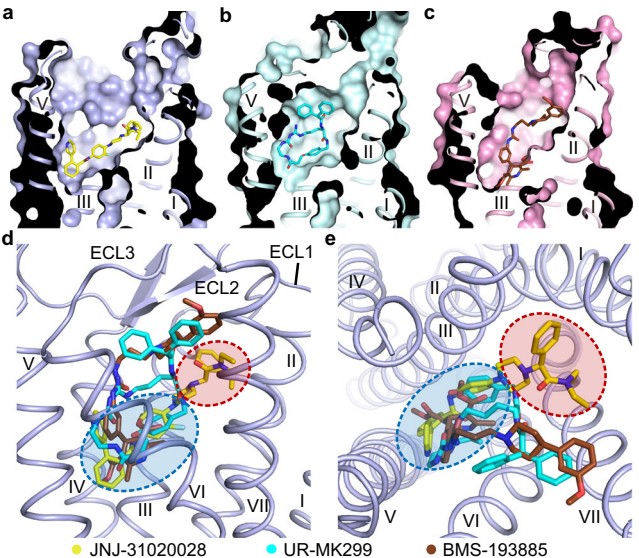

**Fig. 4 Comparison of ligand-binding modes between $Y_2R$ and $Y_1R$.**
**a–c** Cutaway view of ligand-binding pockets in $Y_2R$ and $Y_1R$. The structures of $Y_2R$–JNJ-31020028 (**a**), $Y_1R$–UR-MK299 (**b**) (PDB code: 5ZBQ), and $Y_1R$–BMS-193885 (**c**) (PDB code: 5ZBH) are shown in cartoon and surface representations, with the receptors colored light blue, light cyan, and pink, respectively. The ligands JNJ-31020028, UR-MK299, and BMS-193885 are shown as yellow, cyan, and brown sticks, respectively. **d**, **e** Comparison of ligand-binding sites in $Y_2R$ and $Y_1R$. **d** Side view. **e** Extracellular view. Only the receptor in the $Y_2R$–JNJ-31020028 structure is shown in cartoon representation for clarity. The red ellipse indicates the extended binding pocket in $Y_2R$ and the blue ellipse indicates the binding site at the bottom of the ligand-binding cavity shared by $Y_1R$ and $Y_2R$.

(Fig. 4d, e). In contrast, the phenylethyl and diethyl amide moieties of JNJ-31020028 in $Y_2R$ approach the extracellular surface of the receptor, interacting with ECL1 and the extracellular tips of helices II, III, and VII, which form an extended binding pocket of $Y_2R$ (Fig. 4d, e).

Sequence alignment of NPY receptors shows that most of the residues in the ligand-binding pocket of the $Y_2R$–JNJ-31020028 structure are conserved among the four receptor subtypes except for $V126^{3.28}$, $L183^{4.60}$, $S223^{5.42}$, $L227^{5.46}$, $H285^{6.52}$, $Q288^{6.55}$, and $T308^{7.36}$, suggesting that these residues may be determinants for ligand selectivity of JNJ-31020028 (Supplementary Fig. 5). The role of these residues in governing ligand selectivity was investigated by mutagenesis studies, in which each of the seven residues was replaced with its counterpart in $Y_1R$. It was observed that four out of the seven $Y_2R$-to-$Y_1R$ swap mutations, $V126^{3.28}N$, $L227^{5.46}Q$, $H285^{6.52}T$, and $Q288^{6.55}N$, decreased the antagonistic activity of JNJ-31020028 by 4–186-fold, supporting the importance of these residues in determining the ligand selectivity (Supplementary Table 2). Among them, the residues at the positions 5.46 and 6.55 have also been implied to be important for selectivity and specificity of the antagonist UR-MK299 in $Y_1R^{13}$, suggesting that these residues may play critical roles in ligand selectivity for different NPY receptors. In the $Y_2R$–JNJ-31020028 structure, both residues locate in the bottom region of the ligand-binding pocket (Fig. 2a). The residue $L227^{5.46}$, together with several other hydrophobic residues, forms a hydrophobic patch to accommodate the fluorophenyl and benzamide groups of JNJ-31020028. The replacement of $L^{5.46}$ with $Q^{5.46}$ in $Y_1R$, $Y_4R$, and $Y_5R$ would impede high-affinity binding of JNJ-31020028 at these receptors by disturbing the hydrophobic patch. $Y_2R$ is the only NPY receptor with a glutamine residue at position 6.55. In $Y_1R$ and $Y_4R$, the residue $N^{6.55}$

with a shorter side chain likely weakens the key polar contacts with the benzamide and pyridine groups of the ligand, and probably mediates selectivity. This is reflected by a 186-fold reduction of JNJ-31020028 activity associated with the $Y_2R$ mutation $Q288^{6.55}N$, which represents the most profound effect on the antagonistic activity among the $Y_2R$ mutants we tested (Supplementary Table 2). Instead of a bulky histidine, the residue at position 6.52 is threonine in $Y_1R$, which excludes the polar and hydrophobic interactions between this residue and the benzamide group in JNJ-31020028 and thus may reduce binding affinity. This is supported by the 20-fold drop of JNJ-31020028 activity for the $Y_2R$ mutant $H285^{6.52}T$ (Fig. 3a, Supplementary Fig. 4n, and Supplementary Table 2). In addition to the residues in the bottom region of the ligand-binding pocket, the residues in the extended binding pocket of $Y_2R$ may also account for ligand selectivity. The residue at position 3.28 displays high diversity in NPY receptors ($Y_1R$, $N^{3.28}$; $Y_2R$, $V^{3.28}$; $Y_4R$, $S^{3.28}$; $Y_5R$, $M^{3.28}$). The $Y_2R$ residue $V^{3.28}$ makes a hydrophobic contact with the phenyl ring in the phenylethyl group of JNJ-31020028. The hydrophilic counterparts in the other receptors would impair the hydrophobic interaction and may decrease the binding affinity. In contrast to the high selectivity of the antagonists at different NPY receptors, these receptors recognize the same set of peptide agonists. This may due to conformational flexibility of the peptides, which may adopt distinct interaction patterns with different receptors by adjusting their conformations.

Through comparison of the effects of the key residues within the ligand-binding pocket for different antagonists, our mutagenesis studies revealed two regions in $Y_2R$ that may play different roles in the crosstalk between the receptor and ligand. The subpocket that locates at the bottom of the $Y_2R$ ligand-binding cavity is composed of residues from helices III–VI (Fig. 4d, e), which exhibited a large effect on the inhibitory activity of different antagonists with diverse structures in the NPY-induced IP accumulation assay. This binding site is also shared by $Y_1R$, with mutations at key positions such as 5.46, 6.48, 6.52, and 6.55 substantially impairing the antagonistic activity of several small-molecule antagonists[13]. These findings suggest that different YR antagonists may modulate the activity of their receptors in a similar manner. The interactions between the antagonists and the receptor residues in the bottom region of the ligand-binding pocket may stabilize the receptor inactive conformation and/or block the conformational change that is required for receptor activation. In contrast to the similar behavior of the antagonists in this region, the upper part of the ligand-binding pocket in $Y_2R$ bordered by helices II, III, and VII shows diversity in recognition of different antagonists. Mutations at positions 2.64, 3.28, 3.32, and 7.35 displayed distinct effects on the inhibition of different antagonists in the IP production assay. All these residues are located within the extended binding pocket of $Y_2R$, which is not involved in ligand interaction in the antagonist-bound $Y_1R$ structures[13] (Fig. 4d, e). These results suggest that the extended binding pocket in $Y_2R$ may play a role in selective recognition of various antagonists and could serve as a target site for design of highly selective drugs.

In summary, the $Y_2R$–JNJ-31020028 structure, together with the extensive mutagenesis studies, provides molecule details of $Y_2R$ in recognition of various antagonists and reveals key determinants of ligand selectivity and receptor activation modulation. These findings extend our knowledge about ligand recognition of the NPY receptor family and would facilitate rational drug design targeting different NPY receptors.

## Methods
**Protein engineering and expression of $Y_2R$.** To enable receptor expression and purification, the WT human $Y_2R$ gene (Genewiz) was cloned into a modified

pFastbac1 vector (Invitrogen) with a hemagglutinin signal sequence at the N terminus, and a PreScission protease site followed by a 10 × His-tag and a Flag tag at the C terminus. To improve protein yield and stability, two mutations H149$^{3.51}$Y and S280$^{6.47}$C were introduced into Y$_2$R and 28 amino acids (residues S354-V381) at the C terminus of Y$_2$R were truncated using standard QuikChange PCR. To facilitate crystallization, residues 2–161 of a modified T4L were fused to the receptor N terminus, and residues S251-N256 in ICL3 were replaced by residues 2–148 of a modified flavodoxin (P2A, Y98W)[27] through overlap extension PCR. Sequences of all primers used in this work are shown in Supplementary Table 3.

High-titer recombinant baculovirus (>10$^8$ viral particles per ml) of the modified Y$_2$R was prepared using the Bac-to-Bac Baculovirus Expression System (Invitrogen). *Spodoptera frugiperda* (*Sf*9) insect cells (Invitrogen) were grown to a density of 2 × 10$^6$ cells ml$^{-1}$ in ESF 921 serum-free medium (Expression Systems) at 27 °C and then infected with the viral stock at a multiplicity of infection of 5. The cell pellets were harvested by centrifugation 48 h post infection and stored at −80 °C until use.

**Purification of Y$_2$R–JNJ-31020028 protein.** The cells were disrupted by thawing the frozen cell pellets on ice and then performing dounce homogenization in a hypotonic buffer containing 10 mM HEPES, pH 7.5, 10 mM MgCl$_2$, and 20 mM KCl supplemented with EDTA-free protease inhibitor cocktail (Roche) at the ratio of 1 tablet per 100 ml buffer. Cell membranes were collected by centrifugation at 160,000 × g for 30 min, and washed by repeating dounce homogenization and centrifugation twice in a high osmotic buffer containing 10 mM HEPES, pH 7.5, 10 mM MgCl$_2$, 20 mM KCl, 1 M NaCl, and EDTA-free protease inhibitor cocktail. After that, the high concentration of NaCl was removed by one more wash in the hypotonic buffer. The purified membranes were then resuspended in a buffer containing 10 mM HEPES, pH 7.5, 10 mM MgCl$_2$, 20 mM KCl, 30% (v/v) glycerol and incubated at 4 °C for 1 h in the presence of 50 μM JNJ-31020028 (MedKoo Biosciences), 2 mg ml$^{-1}$ iodoacetamide (Sigma-Aldrich), and EDTA-free protease inhibitor cocktail at the ratio of 1 tablet per 50 ml buffer.

After incubation, the membranes were then solubilized in a buffer containing 50 mM HEPES, pH 7.5, 300 mM NaCl, 0.5% (w/v) n-dodecyl-β-D-maltopyranoside (DDM, Anatrace), and 0.1% (w/v) cholesterol hemisuccinate (CHS, Sigma) at 4 °C for 3 h. Centrifugation at 160,000 × g for 30 min was performed to remove the membrane debris. The supernatant was incubated with TALON IMAC resin (Clontech) at 4 °C overnight in the presence of 10 mM imidazole, pH 7.4. To purify the protein, 25 column volumes of wash buffer (25 mM HEPES, pH 7.5, 300 mM NaCl, 0.05% (w/v) DDM, 0.01% (w/v) CHS, 30 mM imidazole, 10% (v/v) glycerol, and 50 μM JNJ-31020028) were used to remove unspecific binding proteins, and 5 column volumes of elution buffer (25 mM HEPES, pH 7.5, 300 mM NaCl, 0.05% (w/v) DDM, 0.01% (w/v) CHS, 300 mM imidazole, 10% (v/v) glycerol, and 50 μM JNJ-31020028) were used to elute the Y$_2$R–JNJ-31020028 protein. A PD MiniTrap G-25 column (GE Healthcare) was further used to remove imidazole in the elution buffer and decrease the DDM concentration to 0.025% (w/v). The C-terminal 10 × His-tag and glycosylation of the receptor were removed by treating the protein with His-tagged PreScission protease (custom-made) and His-tagged PNGase F (custom-made) at 4 °C overnight. The PreScission protease, PNGase F, and cleaved His-tag were removed by incubating with Ni–NTA superflow resin (Qiagen) at 4 °C for 1 h. The purified Y$_2$R–JNJ-31020028 complex protein was concentrated to about 30 mg ml$^{-1}$ using a 100 kDa molecular-weight cutoff Vivaspin concentrator (Sartorius Stedim Biotech). The protein purity, monodispersity, and thermal stability were analyzed by SDS–PAGE, analytical size-exclusion chromatography, and carry eclipse microscale fluorescence assays.

**Crystallization of Y$_2$R–JNJ-31020028 protein.** To reconstitute the purified protein into the lipidic cubic phase (LCP), the molten lipid that consists of monoolein (Anatrace) and cholesterol (Sigma-Aldrich) at a ratio of 9:1 (w/w) was mixed with the protein solution at a ratio of 2:3 (w/w) using the canonical syringe mixer[37]. To perform crystallization trials, 40 nl of the protein-lipid LCP mixture and 800 nl precipitant solution were dispensed onto 96-well glass sandwich plates (Shanghai FAstal BioTech) using an automatic Gryphon robot (Art-Robbins). The plates were then incubated and imaged in an automated incubator/imager (Rock Imager, Formulatrix) at 20 °C. The diffraction-quality crystals of Y$_2$R–JNJ-31020028 appeared after 3–4 days in 0.1 M HEPES, pH 7.0-7.5, 250–350 mM (NH$_4$)$_2$SO$_4$, and 20-30% PEG500DME, or 0.1 M MES, pH 6.0–6.5, 380–420 mM NH$_4$ tartrate, and 24-26% PEG500DME. The crystals reached to the maximum size of 130 × 30 × 5 μm$^3$ in 1 month. The crystals were harvested by using 50–100 μm microloops (M2-L19-20/50, MiTeGen) and flash frozen in liquid nitrogen.

**Data collection and structure determination.** The X-ray diffraction data of Y$_2$R–JNJ-31020028 crystals were collected at the SPring-8 beamline 41XU, Hyogo, Japan, using a 10 μm × 8 μm minibeam for 0.2 s and 0.2° oscillation per frame with a Pilatus3 6M detector at a wavelength of 1.0000 Å. The structure of Y$_2$R–JNJ-31020028 complex was determined at 2.8 Å resolution by integrating, scaling, and merging the diffraction data from 52 best-diffracting crystals using HKL2000[38]. Molecular replacement was then performed with Phenix[39] using Y$_1$R (PDB code: 5ZBQ)[13], T4L (PDB code: 5KHZ)[40], and flavodoxin (PDB code: 5YOB)[41] as search

models. The T4L fusion protein at the receptor N terminus was not refined due to poor electronic densities. Each asymmetric unit contains one Y$_2$R-flavodoxin molecule. The structure was refined using COOT[42], Phenix,[39] and Buster[43] based on both 2|$F$| − |$F_c$| and |$F_o$| − |$F_c$| maps. According to the Ramachandran plot analysis, the structure was well refined with 100% of the residues in favorable (96.3%) or allowed (3.7%) regions. The final model of Y$_2$R–JNJ-31020028 complex contains 290 residues (E48-V250 and D257-E343) of the 381 residues of Y$_2$R and residues A2-I148 of flavodoxin.

**IP accumulation assay.** The plasmid hY$_2$R_eYFP_pVitro2 encoding the WT Y$_2$R and a fluorophore for investigating the receptor localization was used as template for the generation of receptor point mutations as previously described[16]. Mutagenesis PCR was performed using the Phusion High-Fidelity Polymerase (Thermo Fisher Scientific, Waltham, MA, USA). The plasmids were amplified in chemically competent *E.coli* DH5α (NEB, Ipswich, MA, USA) and the success of the mutagenesis was confirmed by Sanger dideoxy sequencing. The preparations of the plasmid DNA were performed using the Wizard plus Mini or Midi DNA purification kit (Promega, Madison, WI, USA).

HEK293 cells (DMSZ, Braunschweig, Germany) were cultivated in Dulbecco's Modified Eagle Medium (with 4.5 g l$^{-1}$ glucose and L-glutamine) and Ham's F12 (1:1) supplemented with 15% heat-inactivated fetal calf serum at 37 °C and 5% CO$_2$ in humidified atmosphere. Membrane localization of the receptor mutants was observed utilizing the C-terminally tagged fluorophore. HEK293 cells were seeded into eight-well μ slides (IBIDItreat, Martinsried, Germany) and grown to a confluency of 70–80%. The cells were transfected with 1000 ng plasmid DNA using Lipofectamine 2000 (Invitrogen, Carlsbad, CA, USA) according to the manufacturer's protocol (1 μl Lipofectamine per 1 μg DNA). The following day, medium was replaced by 200 μl Opti-MEM (Gibco Thermo Fisher Scientific Inc., Waltham, MA, USA) and 0.5 ng Hoechst 33342 nuclear dye (Sigma-Aldrich, St. Louis, MO, USA) and the cells were starved at 37 °C for 30 min. The membrane localization of the receptor constructs was observed by live cell imaging using an Axio Observer Z1 microscope with an ApoTome.2 Imaging system (Zeiss, Oberkochen, Germany; GFP filter no. 38: ex. 470/40 nm, em. 525/50 nm; DAPI filter no. 02: ex. 365 nm, em. 420 nm).

To determine the contribution of residues to antagonist binding, IP-One accumulation assays (Cisbio, Codolet, France) were performed. HEK293 cells were seeded into six-well plates and transiently cotransfected with 3200 ng of receptor plasmid DNA and 800 ng of the chimeric G protein (Gα$_{\Delta 6qi4myr}$) to redirect the Gα$_i$ signaling pathway to the Gα$_q$ phospholipase C pathway. Transient transfection was carried out using Metafectene as transfection reagent according to the manufacturer's protocol (Biontex, Munich, Germany) over night. The cells were then reseeded into white 384-well microplates (Greiner Bio-One, Kremsmünster, Austria) at a density of 15,000 cells per well and grown for additional 24 h.

For the investigation of different receptor mutants in the binding pocket, the medium was removed and the cells were preincubated with or without 1 μM of the antagonist [JNJ-31020028 or BIIE0246 from Tocris Biosciences (Bristol, UK) or compound 6 (provided by the Bayer Company)], respectively, for 10 min. All dilutions were performed in HBSS containing 20 mM LiCl (stimulation buffer) and DMSO with a final concentration of 0.01%. NPY was added to a final concentration of 1 × 10$^{-12}$ to 1 × 10$^{-5}$ M, and cells were stimulated at 37 °C and 5% CO$_2$ in humidified atmosphere for 90 min.

For the determination of the dose-dependent inhibition of the antagonist in NPY signaling at the WT Y$_2$R versus the crystallization construct (without flavodoxin in ICL3 to enable G protein interaction), the medium was removed, and the cells were preincubated with different concentrations of JNJ-31020028 in stimulation buffer (concentration range 1 × 10$^{-10}$ to 1 × 10$^{-5}$ M) for 10 min. NPY was then added to a final concentration of 0.3 nM, which corresponds to approximately the EC$_{80}$ of the NPY-induced signaling at the Y$_2$R constructs in the absence of antagonist. The cells were stimulated at 37 °C and 5% CO$_2$ in humidified atmosphere for 90 min. The exact value of the effective concentration (EC$_x$) at the day of the assay was determined by control curves in the absence of antagonist.

To determine the amount of cellular IPs, the FRET acceptor (IP1-d2) and the FRET donor (anti-IP1-Cryptate) were reconstituted and diluted in lysis and detection buffer (1:20) as described in the manufacturer's protocol. Three microliters of both solutions were added, and the cells were incubated on a tumbler for 60 min. Fluorescence emission was detected at 620 nm (10 nm bandwidth) and 665 nm (8 nm bandwidth) at the plate reader (Tecan Spark, Tecan, Männedorf, Switzerland). The HTRF ratio was calculated by dividing the detected emission at 665 nm by the detected emission at 620 nm. The EC$_{50}$ and EC$_{50}$ ratios (fixed concentration of antagonist) as well as IC$_{50}$ (dose-dependent inhibition of the antagonist, fixed NPY concentration) were calculated by three-parameter logistic fit using the GraphPad Prism 5 software (GraphPad Software, San Diego, CA, USA).

**Binding assay.** The binding affinity of JNJ-31020028 to the engineered Y$_2$R construct for crystallization was verified by NanoBRET binding assays. For this purpose, a Nanoluciferase[44] (Promega, Madison, WO, USA) followed by a flexible Ser-Gly$_4$-Ser-Linker was genetically fused to the N terminus of the receptor in an Y$_2$-eYFP_N1 vector[45]. The Nanoluciferase replaced the similarly sized N-terminal T4L fusion in the crystallization construct. To facilitate correct expression of this large N-terminal domain and export to the plasma membrane, a signal sequence of

human interleukin 6 (MNSFSTSAFGPVAFSLGLLLVLPAAFPAP) was used[44]. HEK293 cells were transiently transfected with 8 µg of the corresponding vectors per T75 flask using MetafectenePro (Biontex) following the manufacturer's instructions. Twenty-four hours post transfection, the cells were harvested in PBS ($-Ca^{2+}$; $-Mg^{2+}$) and frozen ($-80\,°C$). The membranes were prepared as described previously[34]. Briefly, the cells were lysed in a hypotonic TRIS buffer (50 mM Tris-HCl, pH 7.5 and 50 µM Pefabloc SC) and homogenized using a manual dounce homogenizer for 15 times. Nuclei and cell debris was removed by centrifugation at $820 \times g$ for 10 min (4 °C), and the microsomal membranes in the supernatant were collected by centrifugation at $20{,}000 \times g$ for 60 min (4 °C). The resulting membrane pellet was homogenized for 15 times again in a buffer containing 25 mM HEPES, pH 7.4, 25 mM $CaCl_2$, 1 mM $MgCl_2$, and 50 µM Pefabloc SC, and then recentrifuged at $20{,}000 \times g$ for 90 min (4 °C). The total protein amount was determined by a Bradford assay. The membrane preparations were stored in aliquots at $-80\,°C$.

Tetramethylrhodamine (TAMRA)-labeled NPY was used as labeled species in the NanoBRET-based binding assays. The fluorescence label was introduced at position 18 of the peptide by solid-phase peptide synthesis, using a Lys with orthogonal side chain protection group replacing the native Ala residue. Selective deprotection of the Lys(Dde) and coupling of the TAMRA fluorophore were performed as described[46], and the peptide was purified to >95%. The native-like activity of this peptide was verified in IP accumulation assays ($EC_{50} = 0.03$ nM, $pEC_{50} = 10.47 \pm 0.09$; compared to NPY $pEC_{50} = 10.30 \pm 0.13$).

Membranes containing 0.25 µg total protein were used for each data point, which corresponded to a total luminescence of ~300,000 RLU (430-470 nm filter; 4 µM coelenterazine H). The membranes were incubated with 300 nM K18 (TAMRA)-NPY and varying concentrations of the antagonist for 90 min in HBSS buffer containing 25 mM HEPES, pH 7.4, 0.1% bovine serum albumin, and 50 µM Pefabloc SC protease inhibitor in a total volume of 100 µl in solid black 96-well plates. Directly before the measurement, 10 µl coelenterazine H in HBSS/HEPES was added to a final concentration of 4 µM. BRET was measured in a Tecan Spark plate reader (Tecan, Männedorf, Switzerland) with the following filter sets: luminescence (L) 430–470 nm and fluorescence (F) 550–700 nm. The BRET ratio was calculated by the ratio of F/L, and corrected by the background values. The $IC_{50}$ values were determined using a three-parameter logistic fit in the GraphPad Prism 5 software (GraphPad Software, San Diego, CA, USA).

**Reporting summary**. Further information on research design is available in the Nature Research Reporting Summary linked to this article.

## Data availability
Atomic coordinates and structure factors of the $Y_2R$–JNJ-31020028 structure have been deposited in the Protein Data Bank under accession code 7DDZ [https://doi.org/10.2210/pdb7DDZ/pdb]. All relevant data are available from the corresponding authors upon reasonable request. Source data are provided with this paper.

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

## Acknowledgements

This work was supported by the National Science Foundation of China grants 31825010 (B.W.) and 31700653, the National Key R&D Program of China 2018YFA0507000 (Q.Z. and B.W.), the German Research Foundation (421152132, CRC 1423, A04 to A.G.B.-S. and B03 to A.K.), CAS Strategic Priority Research Program XDB37030100 (B.W. and Q.Z.), and the Shanghai Science and Technology Committee grants 19JC1416200 (B.W.) and 18XD1404800 (Q.Z.). The synchrotron radiation experiments were performed at the BL41XU of SPring-8 with approval of the Japan Synchrotron Radiation Research Institute (Proposal nos 2019A2543, 2019B2543, 2019A2541, and 2019B2541). We thank the beamline staff members K. Hasegawa, N. Mizuno, T. Kawamura, and H. Murakami of the BL41XU for help with X-ray data collection, and K. Löbner for help in cell microscopy and tissue culture. We thank M. Keller for providing $Y_2R$ ligands for initial screening.

## Author contributions

T.T. optimized the construct, purified the $Y_2R$ protein, performed crystallization trials, solved the structure, and helped with manuscript preparation. C.H. performed mutagenesis and functional studies, and helped with manuscript preparation. Q.C., W.Z., and H.Q. helped with protein sample optimization. A.K. helped with selection of mutants, performed functional studies and binding assays, and edited the manuscript. H.Z. collected X-ray diffraction data. X.Z. and S.H. helped with structure determination. C.Y. and L.M. expressed the protein. A.B.-S. oversaw the mutagenesis and functional studies, and edited the manuscript. B.W. and Q.Z. initiated the project, planned and analyzed experiments, supervised the research, and wrote the manuscript with input from all coauthors.

## Competing interests

The authors declare no competing interests.
