## [Peer Review File · Nature Communications]

REVIEWER COMMENTS

Reviewer #1 (Remarks to the Author):

This manuscript by Tang et al reports the structural and pharmacological characterization of the action of antagonists on the human NPY receptor Y2R. NPY receptors are interesting neuropeptide receptors in drug development due to their roles in the stimulation of food intake and the modulation of multiple functional aspects of the CNS. The authors solved a crystal structure of Y2R bound to the antagonist JNJ-31020028 and performed extensive mutagenesis studies to identify critical residues in the binding of three chemically distinct antagonists. The authors also discussed potential structural determinants in the lower and upper regions of the ligand-binding pocket for the ligand selectivity of NPY receptors. The paper is well written and the data presentation is clear.

The same group(s) reported the crystal structures of antagonist-bound Y1R in 2018, which revealed different patterns of interactions for the two Y1R antagonists and insights into the binding mode of NPY. In the current study, although the structural determination of another NPY receptor represents a technical breakthrough, the results are rather limited in advancing our understanding of the signaling and pharmacology of NPY receptors:

1. The overall structure of Y2R is very similar to other GPCRs for neuropeptides especially Y1R. The only noticeable differences are the slight movement of the extracellular ends of TM2 and TM6 as pointed out by the authors. The ligand-binding pocket also overlaps with those in other neuropeptide GPCRs. This might not be a major issue if the structural insights could inform the therapeutic potential and development of Y2R-targeting ligands. The authors discussed the molecular basis for the Y2R-selective action of JNJ-31020028 and the different modes of action of JNJ-31020028, BIIE0246, and Compound 6. However, it is not clear why the authors chose those antagonists in their studies. Are they associated with different pharmacological properties that may lead to their differences in therapeutic applications? Also, ligand selectivity seems not to be the major reason why previous drug development efforts on Y2R didn't succeed.
2. It would be more informative if the authors have determined the crystal structures of Y2R with other ligands, similar to what they did in their structural studies on Y1R, other than just describe residues that may interact with them. In particular, BIIE0246 is the most widely used pharmacological tool for Y2R. The actual structural comparison of different binding modes of Y2R antagonists will significantly increase the research impact. It is possible that the extracellular region may adopt different conformations for distinct antagonists.
3. The authors proposed that the interactions with residues Q3.32 and W6.48 are important for the antagonistic action of JNJ-31020028. However, discussion of potential mechanisms for NPY binding and receptor activation is lacking. Y1R and Y2R differ in the recognition of NPY. It would be more informative if the authors could speculate on the agonist peptide recognition by Y2 and how is that different from Y1R.

Other technique concerns and comments:

4. The authors used a modified Y2R construct to obtain the crystal structure. They only provided the NPY-induced IP1 accumulation data to validate the functionality of such a construct, which is not sufficient. Proof of unchanged binding of antagonists is needed since the entire paper is focused on the binding of antagonists. This is particularly important considering the fact that the mutation site S280C is close to the antagonist-binding pocket.
5. The authors discussed residues that are important for the selectivity of JNJ-31020028 for Y2R over other NPY receptors. Some supporting mutagenesis data on other NPY receptors may be helpful to validate the structural insights, e.g. gain-of-function of mutations in Y1R for the binding of JNJ-

31020028. Or the authors could discuss the potential binding modes of non-selective NPY receptor antagonists? If there is none, speculation on why Y1R and Y2R recognize the same set of peptide agonists but distinct antagonists would be informative since non-selective NPY antagonists may provide some therapeutic advantages over selective antagonists (PMID 20972986).

6. An explanation for the JNJ stereoisomer selectivity would be helpful.

7. It is important to provide Ramachandran analysis results for the validation of structure refinement (Supplementary Table 1).

Reviewer #2 (Remarks to the Author):

The paper by Tang et al describes the crystal structure of the Y2 receptor bound to the small-molecule antagonist JNJ-31020028. Furthermore, the paper includes extensive mutagenesis studies of the Y2 receptor and a comparison with the structure of the Y1 receptor bound to another small-molecule antagonist.

The human Y receptors are ubiquitous. The natural ligands for the Y receptors are peptides, including NPY and PYY 3-36, that have a high degree of Y receptor subtype selectivity. These receptors have been proposed as important drug targets. The currently most promising drug candidates are probably peptide agonists with a high selectivity for the Y2 receptor, which are biopharmaceutical drug candidates for the treatment of obesity, when administered together with GLP-1 analogs.

The crystal structure of the Y2 receptor bound with a small-molecule antagonist is very relevant to the field. The combination with mutagenesis studies is particularly strong and lends further support for the interpretations. Based on this, the authors are able to rationalize already reported SAR data.

This reviewer has no major objection to this manuscript. The reported Y2 structure will be of interested to many medicinal chemists but also to structural biologists. The work is state-of-the-art and the manuscript is well written.

Minor mistake:

Line 29: 'treat' should be 'treatment'

Responses to the reviewers' comments

Reviewer #1 (Remarks to the Author):

This manuscript by Tang et al reports the structural and pharmacological characterization of the action of antagonists on the human NPY receptor Y₂R. NPY receptors are interesting neuropeptide receptors in drug development due to their roles in the stimulation of food intake and the modulation of multiple functional aspects of the CNS. The authors solved a crystal structure of Y₂R bound to the antagonist JNJ-31020028 and performed extensive mutagenesis studies to identify critical residues in the binding of three chemically distinct antagonists. The authors also discussed potential structural determinants in the lower and upper regions of the ligand-binding pocket for the ligand selectivity of NPY receptors. The paper is well written and the data presentation is clear.

The same group(s) reported the crystal structures of antagonist-bound Y₁R in 2018, which revealed different patterns of interactions for the two Y₁R antagonists and insights into the binding mode of NPY. In the current study, although the structural determination of another NPY receptor represents a technical breakthrough, the results are rather limited in advancing our understanding of the signaling and pharmacology of NPY receptors:

1. The overall structure of Y₂R is very similar to other GPCRs for neuropeptides especially Y₁R. The only noticeable differences are the slight movement of the extracellular ends of TM2 and TM6 as pointed out by the authors. The ligand-binding pocket also overlaps with those in other neuropeptide GPCRs. This might not be a major issue if the structural insights could inform the therapeutic potential and development of Y₂R-targeting ligands. The authors discussed the molecular basis for the Y₂R-selective action of JNJ-31020028 and the different modes of action of JNJ-31020028, BIIE0246, and Compound 6. However, it is not clear why the authors chose those antagonists in their studies. Are they associated with different pharmacological properties that may lead to their differences in therapeutic applications? Also, ligand selectivity seems not to be the major reason why previous drug development efforts on Y₂R didn't succeed.

— We thank the reviewer for this comment. The compounds JNJ-31020028, BIIE0246, and Compound 6 were chosen as they are all selective Y₂R antagonists but differ in molecular size, chemical structure, and antagonistic activity (Supplementary Table 2). BIIE0246 was the first highly active and selective Y₂R antagonist, but its clinical effectiveness is hampered by the poor blood-brain-barrier permeability. In addition, some off-target effects were later discovered (Brothers, S. P. *et al.*, *Mol. Pharmacol.* 77:46–57, 2010). In contrast, a new series of Y₂R-specific antagonists (Brothers, S. P. *et al.*, *Mol. Pharmacol.* 77:46–57, 2010; Mittapalli *et al.*, *Bioorg. Med. Chem. Lett.* 22:3916-20, 2012), including compound 6 have much better pharmacokinetic properties and blood-brain-barrier permeability, but many compounds of this series had comparably little antagonistic activity. JNJ-31020028 combines good pharmacokinetic properties, blood-brain-barrier permeability, and good antagonistic activity. Therefore, studying the binding and antagonistic behavior of these compounds would provide more information on molecular details that govern ligand recognition and antagonistic activity. As discussed in the manuscript (page 13), by comparing the effects of key residues within the ligand-binding pocket on the activity of the three diverse antagonists using mutagenesis, two regions in Y₂R that play distinct roles in the crosstalk between the receptor and ligand were identified. The interactions between the antagonist and the residues at the bottom of the ligand-binding cavity are responsible for stabilizing the receptor inactive conformation and/or block the conformational change required for receptor activation, while the upper part of the ligand-

binding pocket is key for selective recognition of various antagonists. These findings would facilitate future drug development by targeting different site(s) to achieve optimal antagonistic activity and/or high ligand selectivity. To make the above clear in the manuscript, the first sentence in the “Recognition between Y₂R and other antagonists” section (paragraph 2, page 8) has been revised as “To obtain more molecular details that govern ligand recognition and antagonistic activity, which would facilitate future drug discovery, we performed additional mutagenesis studies on two other representative antagonists of Y₂R that differ in size, structure, antagonistic activity, and blood-brain-barrier permeability²⁰, BIIE0246 and Compound 6 (Supplementary Fig. 1g, h)”.

Regarding ligand selectivity leading to the failure of Y₂R drug development, it has been reported that BIIE0246 binds to opioid and adrenergic receptors with submicromolar affinities, as well as to several other receptors with low micromolar affinities, which may result in off-target effects (Brothers, S. P. *et al.*, *Mol. Pharmacol.* 77:46–57, 2010). Thus, ligand selectivity appears to be an issue to limit the clinical usage of some Y₂R antagonists. In addition to selectivity, low potency and poor blood-brain-barrier permeability also hamper the development of Y₂R drugs. To make this clear, the statement in paragraph 2, page 3 has been revised as “A number of Y₂R agonists and antagonists have shown therapeutic potential in the treatment of obesity and anxiety¹⁷⁻¹⁹, but their clinical application has been limited by low potency and selectivity and poor blood-brain-barrier permeability^{12,14,19,20”}.

2. It would be more informative if the authors have determined the crystal structures of Y₂R with other ligands, similar to what they did in their structural studies on Y₁R, other than just describe residues that may interact with them. In particular, BIIE0246 is the most widely used pharmacological tool for Y₂R. The actual structural comparison of different binding modes of Y₂R antagonists will significantly increase the research impact. It is possible that the extracellular region may adopt different conformations for distinct antagonists.

— We appreciate the reviewer’s comment. We did try to crystallize the Y₂R-BIIE0246 complex and obtained some tiny crystals. However, due to the relatively poor protein stability of this complex and its low mobility in lipidic cubic phase compared to the JNJ-31020028-bound receptor, the crystals were hard to be optimized and no diffraction data were obtained (see figure below). Further effort is needed to solve the structure. Indeed, the extracellular region may adopt different conformations for distinct antagonists, given that the upper part of ligand-binding pocket that is adjacent to the extracellular loops exhibits different behaviors when bound to different antagonists (as discussed above). However, more structural information is required to illustrate the role of the extracellular region in mediating ligand selectivity.

- Comparison of Y₂R-JNJ-31020028 and Y₂R-BIIE0246 complexes. **a**, Analytical size-exclusion chromatography (aSEC) of Y₂R (crystallization construct) in complex with JNJ-31020028 or BIIE0246. The peaks for protein aggregation and monomer are indicated by red dashed lines. The results show lower monomer: aggregation ratio of the Y₂R-BIIE0246 complex than the Y₂R-JNJ-31020028 complex, indicating that the BIIE0246-bound receptor has worse protein stability. **b**, Crystal image of the Y₂R-BIIE0246 complex. The crystal size is about 10 μm. **c** and **d**, Fluorescence recovery after photobleaching curves of Y₂R in complex with BIIE0246 (**c**) or JNJ-31020028 (**d**) in lipidic cubic phase. The results show higher recovery rate of Y₂R-JNJ-31020028 compared to Y₂R-BIIE0246, indicating better mobility of the receptor when bound to JNJ-31020028.

3. The authors proposed that the interactions with residues Q3.32 and W6.48 are important for the antagonistic action of JNJ-31020028. However, discussion of potential mechanisms for NPY binding and receptor activation is lacking. Y_1R and Y_2R differ in the recognition of NPY. It would be more informative if the authors could speculate on the agonist peptide recognition by Y_2 and how is that different from Y_1R .

— Thank the reviewer for this comment. Indeed, although both Y_1R and Y_2R bind to NPY with high affinity, these two receptors interact with the peptide agonist through different patterns, supported by previous mutagenesis studies. The variation occurs in both the peptide N- and C-termini, such as distinct binding partners for the highly conserved NPY receptor residue D^{6.59} (Merten, N. *et al.*, *J. Biol. Chem.* 282:7543-7551, 2007) and different requirements of the peptide N terminus upon binding to different receptor subtypes (Pedragosa-Badia, X. *et al.*, *Front. Endocrinol.* 4:5, 2013; Cabrele, C. *et al.*, *J. Pept. Sci.* 6:97-122, 2000). This has been extensively discussed in many previous publications including our Y_1R structure paper (Yang, Z. *et al.*, *Nature* 556:520-524, 2018). This is truly an important topic, but will need significant extra data to make a clear and accurate enough speculation on it. As this is not the main scope of the present study, we hope the reviewer would agree with us not to include such discussion in the manuscript.

Other technique concerns and comments:

4. The authors used a modified Y_2R construct to obtain the crystal structure. They only provided the NPY-induced IP1 accumulation data to validate the functionality of such a construct, which is not sufficient. Proof of unchanged binding of antagonists is needed since the entire paper is focused on the binding of antagonists. This is particularly important considering the fact that the mutation site S280C is close to the antagonist-binding pocket.

— The suggestion is well taken. We assessed the antagonist binding by performing a NanoBRET-based binding assay. The data showed a comparable binding affinity of JNJ-31020028 to the wild-type Y_2R and crystallization construct (removing the N-terminal T4L fusion to ensure suitable distance for BRET). Furthermore, additional IP accumulation assays verified that the antagonist activity of JNJ-31020028 at the crystallization construct (removing the ICL3-flavodoxin fusion to allow G protein coupling) is indistinguishable from the wild type. These data support the unchanged binding of the antagonist. The new data have been added to Supplementary Figure 1 (see figure below). Experimental details of the binding assay have been added in the Methods. The statement in the “Structure determination of Y_2R -JNJ-31020028 complex” section (paragraph 1, page 4) has been revised as “Functional assays indicate that these modifications have little effect on binding and antagonistic activity of JNJ-31020028 and receptor signaling (Supplementary Fig. 1a, b)”.

• **Supplementary Figure 1. Function validation of crystallization construct.** **a**, NPY-induced IP accumulation of Y_2R and inhibition by JNJ-31020028. “Construct_no flavodoxin” indicates the modified Y_2R protein used for crystallization except that the ICL3-flavodoxin fusion was removed to allow G protein coupling. Data are shown as mean \pm SEM from at least four independent experiments performed in triplicate. See Supplementary Table 2 for detailed statistical evaluation. **b**, NanoBRET-based binding assay of wild-type Y_2R (WT) and the modified Y_2R protein. “Construct_no T4L” indicates the modified Y_2R

protein used for crystallization except that the N-terminal T4L fusion was removed to ensure suitable distance for BRET. Data are shown as mean \pm SEM from three independent experiments performed in triplicate.

5. The authors discussed residues that are important for the selectivity of JNJ-31020028 for Y_2R over other NPY receptors. Some supporting mutagenesis data on other NPY receptors may be helpful to validate the structural insights, e.g. gain-of-function of mutations in Y_1R for the binding of JNJ-31020028. Or the authors could discuss the potential binding modes of non-selective NPY receptor antagonists? If there is none, speculation on why Y_1R and Y_2R recognize the same set of peptide agonists but distinct antagonists would be informative since non-selective NPY antagonists may provide some therapeutic advantages over selective antagonists (PMID 20972986).

— As suggested, we have performed additional mutagenesis studies to assess the effect of Y_2R -to- Y_1R swap mutations of the seven Y_2R residues that may account for ligand selectivity on the antagonistic activity of JNJ-31020028. A 4–186-fold reduction of antagonistic activity was observed for four of the mutants (V126^{3.28}N, L227^{5.46}Q, H285^{6.52}T, and Q288^{6.55}N) (see figure below), supporting the importance of these residues in determining the JNJ-31020028 selectivity. The new data have been added in Supplementary Table 2.

- IP accumulation of wild-type Y_2R and its Y_2R -to- Y_1R swap mutants induced by NPY (black curves) or NPY with the presence of antagonist JNJ-31020028 (1 μ M) (blue curves). Data are shown as mean \pm SEM from at least three independent experiments performed in triplicate. Detailed statistical evaluation is included in Supplementary Tables 2. **a**, Wild type; **b**, L227^{5.46}Q; **c**, Q288^{6.55}N; **d**, H285^{6.52}T; **e**, V126^{3.28}N; **f**, L183^{4.60}F; **g**, S223^{5.42}L; **h**, F308^{7.36}L.

Based on the new data, the discussion about the selectivity of JNJ-31020028 has been revised as “Sequence alignment of NPY receptors shows that most of the residues in the ligand-binding pocket of the Y_2R -JNJ-31020028 structure are conserved among the four receptor subtypes except for V126^{3.28}, L183^{4.60}, S223^{5.42}, L227^{5.46}, H285^{6.52}, Q288^{6.55}, and T308^{7.36}, suggesting that these residues may be determinants for ligand selectivity of JNJ-31020028 (Supplementary Fig. 5). The role of these residues in governing ligand selectivity was investigated by mutagenesis studies, in which each of the seven residues was replaced with its counterpart in Y_1R . It was observed that four out of the seven Y_2R -to- Y_1R swap mutations, V126^{3.28}N, L227^{5.46}Q, H285^{6.52}T, and Q288^{6.55}N, decreased the antagonistic activity of JNJ-31020028 by 4–186-fold, supporting the importance of these residues in determining the ligand selectivity (Supplementary Table 2). Among them, the residues at the positions 5.46 and 6.55 have also been implied to be important for selectivity and specificity of the antagonist UR-MK299 in Y_1R ¹³, suggesting that these residues may play critical roles in ligand selectivity for different NPY receptors. In the Y_2R -JNJ-31020028 structure, both residues locate in the

bottom region of the ligand-binding pocket (Fig. 2a). The residue L227^{5,46}, together with several other hydrophobic residues, form a hydrophobic patch to accommodate the fluorophenyl and benzamide groups of JNJ-31020028. The replacement of L^{5,46} with Q^{5,46} in Y₁R, Y₄ receptor (Y₄R), and Y₅ receptor (Y₅R) would impede high-affinity binding of JNJ-31020028 at these receptors by disturbing the hydrophobic patch. Y₂R is the only NPY receptor with a glutamine residue at position 6.55. In Y₁R and Y₄R, the residue N^{6,55} with a shorter side chain likely weakens the key polar contacts with the benzamide and pyridine groups of the ligand, and probably mediates selectivity. This is reflected by a 186-fold reduction of JNJ-31020028 activity associated with the Y₂R mutation Q288^{6,55}N, which represents the most profound effect on the antagonistic activity among the Y₂R mutants we tested (Supplementary Table 2). Instead of a bulky histidine, the residue at position 6.52 is threonine in Y₁R, which excludes the polar and hydrophobic interactions between this residue and the benzamide group in JNJ-31020028 and thus may reduce binding affinity. This is supported by the 20-fold drop of JNJ-31020028 activity for the Y₂R mutant H285^{6,52}T (Fig. 3a, Supplementary Fig. 4n, and Supplementary Table 2). In addition to the residues in the bottom region of the ligand-binding pocket, the residues in the extended binding pocket of Y₂R may also account for ligand selectivity. The residue at position 3.28 displays high diversity in NPY receptors (Y₁R, N^{3,28}; Y₂R, V^{3,28}; Y₄R, S^{3,28}; Y₅R, M^{3,28}). The Y₂R residue V^{3,28} makes a hydrophobic contact with the phenyl ring in the phenylethyl group of JNJ-31020028. The hydrophilic counterparts in the other receptors would impair the hydrophobic interaction and may decrease the binding affinity” in pages 11-13.

Regarding the different behaviors of NPY receptors recognizing the same peptide agonists but distinct antagonists, this may be due to high conformational flexibility of the peptide agonists, which may adopt distinct interaction patterns with different receptors by adjusting their conformations. In contrast, the small-molecule antagonists usually have rigid structures and recognize a binding site with specific size and electrostatics. This is supported by our observation that the Y₂R mutation Q288^{6,55}N, where the side chain was shortened by one carbon, caused a 186-fold drop of the antagonistic activity of JNJ-31020028. Based on this, the statement “In contrast to the high selectivity of the antagonists at different NPY receptors, these receptors recognize the same set of peptide agonists. This may due to conformational flexibility of the peptides, which may adopt distinct interaction patterns with different receptors by adjusting their conformations” has been added to paragraph 1, page 13.

6. *An explanation for the JNJ stereoisomer selectivity would be helpful.*

— We apologize for not providing information of the stereoisomer selectivity. Actually, as previously described, the R-isomer and S-isomer of JNJ-31020028 have similar Y₂R affinity (R-isomer, IC₅₀ = 9 ± 2 nM; S-isomer, IC₅₀ = 14 ± 3 nM) (Swanson, D.M. *et al.*, *Bioorg. Med. Chem. Lett.* 21:5552-5556, 2011). This is consistent with the fact that both isomers occupy the same binding site and form similar interactions with the receptor. To make this clear, the description in paragraph 1, page 4 has been revised as “The ligand JNJ-31020028 used in protein purification is a racemic mixture of R-isomer and S-isomer (molar ratio = 1:1), which have similar Y₂R affinity²⁴”.

7. *It is important to provide Ramachandran analysis results for the validation of structure refinement (Supplementary Table 1).*

— We followed the reviewer’s suggestion and have provided Ramachandran analysis results (favored, 96.3%; allowed, 3.7%; disallowed, 0.0%) in Supplementary Table 1.

Reviewer #2 (Remarks to the Author):

The paper by Tang et al describes the crystal structure of the Y₂ receptor bound to the small-molecule antagonist JNJ-31020028. Furthermore, the paper includes extensive mutagenesis studies of the Y₂ receptor and a comparison with the structure of the Y₁ receptor bound to another small-molecule antagonist.

The human Y receptors are ubiquitous. The natural ligands for the Y receptors are peptides, including NPY and PYY 3-36, that have a high degree of Y receptor subtype selectivity. These receptors have been proposed as important drug targets. The currently most promising drug candidates are probably peptide agonists with a high selectivity for the Y₂ receptor, which are biopharmaceutical drug candidates for the treatment of obesity, when administered together with GLP-1 analogs.

The crystal structure of the Y₂ receptor bound with a small-molecule antagonist is very relevant to the field. The combination with mutagenesis studies is particularly strong and lends further support for the interpretations. Based on this, the authors are able to rationalize already reported SAR data.

This reviewer has no major objection to this manuscript. The reported Y₂ structure will be of interested to many medicinal chemists but also to structural biologists. The work is state-of-the-art and the manuscript is well written.

— We are grateful to the reviewer for these comments.

Minor mistake:

Line 29: 'treat' should be 'treatment'

— This has been corrected as suggested.

REVIEWER COMMENTS

Reviewer #1 (Remarks to the Author):

Most of my concerns have been addressed. Additional mutagenesis data and discussion of ligand selectivity are helpful. I still have minor concerns regarding the functional characterization of the Y2R crystallization construct in Supplementary Figure 1. For panel a, after showing that the NPY EC50s are the same for the wt and the engineered receptor, the authors should then determine the dose-dependent inhibition of the antagonist in NPY signaling using a fixed concentration of NPY (e.g. EC50 or EC80), not the dose-dependent stimulation of NPY in the presence of a fixed concentration of the antagonist, to prove the unchanged action of the antagonist. This is basic pharmacology. 2. For panel b, the change of K_i of the JNJ compound is small but significant (if p-test is provided). Although there is no consensus on how much change of ligand affinity is accepted when engineering GPCRs for structural studies, the authors at least need to acknowledge that instead of stating that "have little effect on binding and antagonistic activity of JNJ-31020028". The effect in the NPY action shown in panel a is "little".

Nevertheless, the additional efforts of the authors in improving this manuscript are appreciated. Despite minor issues, I believe the scientific impact and research quality are sufficient for acceptance for publication.

Reviewer #2 (Remarks to the Author):

I appreciate the revisions to this manuscript.

Responses to reviewers' comments

Reviewer #1 (Remarks to the Author):

Most of my concerns have been addressed. Additional mutagenesis data and discussion of ligand selectivity are helpful. I still have minor concerns regarding the functional characterization of the Y2R crystallization construct in Supplementary Figure 1. For panel a, after showing that the NPY EC50s are the same for the wt and the engineered receptor, the authors should then determine the dose-dependent inhibition of the antagonist in NPY signaling using a fixed concentration of NPY (e.g. EC50 or EC80), not the dose-dependent stimulation of NPY in the presence of a fixed concentration of the antagonist, to prove the unchanged action of the antagonist. This is basic pharmacology. 2. For panel b, the change of K_i of the JNJ compound is small but significant (if p -test is provided). Although there is no consensus on how much change of ligand affinity is accepted when engineering GPCRs for structural studies, the authors at least need to acknowledge that instead of stating that "have little effect on binding and antagonistic activity of JNJ-31020028". The effect in the NPY action shown in panel a is "little".

— We thank the reviewer for these comments. As suggested, we have performed additional signaling assay using a fixed concentration of NPY (0.3 nM, \sim EC₈₀). The dose-dependent inhibition of NPY-induced IP accumulation by JNJ-31020028 showed a comparable IC₅₀ for the crystallization construct (removing flavodoxin in ICL3 to enable G protein coupling) to that of the wild-type (WT) receptor (WT: IC₅₀ = 4.8 nM, pIC₅₀ \pm SEM = 7.32 \pm 0.10; construct_no flavodoxin: IC₅₀ = 6.0 nM, pIC₅₀ \pm SEM = 7.22 \pm 0.10). These results further support the unchanged action of the antagonist. The new data have been added to Supplementary Figure 1b (see figure below). The experimental details have been updated in the Methods section.

• **Supplementary Figure 1. Function validation of crystallization construct. b.** Inhibition of NPY-induced IP accumulation of Y₂R by JNJ-31020028. A fixed concentration of NPY (0.3 nM, \sim EC₈₀) was used to stimulate IP accumulation. Data are shown as mean \pm SEM from three independent experiments performed in triplicate (WT: IC₅₀ = 4.8 nM, pIC₅₀ \pm SEM = 7.32 \pm 0.10; construct_no flavodoxin: IC₅₀ = 6.0 nM, pIC₅₀ \pm SEM = 7.22 \pm 0.10). Data are normalized to the actual effective concentration (EC_x) at the day of the assay.

Regarding the change of ligand affinity, we calculated the P value (one-way ANOVA followed by Dunnett's posttest, compared with the response of WT) for the pIC₅₀ obtained from the NanoBRET-based binding assay (see table below), and found no significant difference between the WT and construct. Thus, it was confirmed that the receptor modifications have little effect on JNJ-31020028 binding. The IC₅₀ values have been added to the legend of Supplementary Figure 1.

	IC ₅₀ (nM)	pIC ₅₀		n
		Mean \pm SEM	P	
WT	1.3	8.89 \pm 0.08	/	3
Construct_no T4L	0.7	9.17 \pm 0.13	0.557	3

Nevertheless, the additional efforts of the authors in improving this manuscript are appreciated. Despite minor issues, I believe the scientific impact and research quality are sufficient for acceptance for publication.

— We are grateful to the reviewer for this comment.

Reviewer #2 (Remarks to the Author):

I appreciate the revisions to this manuscript.

— We are grateful to the reviewer for this comment.